# Influence of Copper and Zinc Tailing Powder on the Hydration of Composite Cementitious Materials

**DOI:** 10.3390/ma15165612

**Published:** 2022-08-16

**Authors:** Weiwei Han, Fanghui Han, Ke Zhang

**Affiliations:** 1Beijing Key Laboratory of Urban Underground Space Engineering, Department of Civil Engineering, University of Science and Technology Beijing, Beijing 100083, China; 2Jiangsu Key Laboratory of Construction Materials, School of Materials Science and Engineering, Southeast University, Nanjing 211189, China; 3China Construction Third Engineering Bureau Group Co., Ltd., Wuhan 430074, China

**Keywords:** copper and zinc tailing powder, composite cementitious materials, hydration heat, strength development, microstructure

## Abstract

Copper and zinc tailing powder (CZTP) is finely ground waste after copper minerals and zinc minerals have been extracted from ores during beneficiation. CZTP has certain potential cementitious properties and can be used in composite cementitious materials. The pore size distribution and hydrate phase assemblage of the hardened samples are investigated using MIP and XRD. SEM is employed to examine the microstructure of the specimens. The chemically bonded water is used to measure the degree of hydration. CZTP lowers the hydration heat evolution rate and the total hydration heat. The hydration heat evolution rate reduces as the w/b ratio rises, whereas the total hydration heat of blended cement paste rises. CZTP diminishes the strength development of the Portland-CZTP system, and the strength decreases as the CZTP level increases. CZTP reduces the critical pore diameters of the Portland-CZTP system with w/b = 0.3 after curing for 3 d and 28 d, while increasing the critical pore diameters of samples with w/b = 0.45 at the same age. CZTP increases the gel micropores of Portland-CZTP. Although CZTP increases the pore volume content of blended cement pastes with w/b = 0.3, the volume of harmful pores decreases. The pore volume content of the Portland-CZTP system decreases as the w/b ratio increases. However, the volume of harmful pores increases with a higher w/b ratio. The main hydration products in the Portland-CZTP system are portlandite, ettringite, and C-S-H. CZTP mainly played the role of filling or acting as a microaggregate in the Portland-CZTP system.

## 1. Introduction

Portland cement is the largest and most widely used cement at present, and it can not only be used in civil facilities but also meet the needs of some special engineering [1]. However, the production of Portland cement consumes a large amount of energy and releases a tremendous amount of carbon dioxide, breaking down the balance of the greenhouse gas equation in the atmosphere. Some studies have shown that approximately 0.87 tons of carbon dioxide will be released to produce one ton of Portland cement [2]. To reduce the pressures on our environment, cementitious materials are usually prepared using conventional cement systems based on partial replacement of ordinary Portland cement with mineral admixtures such as pulverized fly ash [3], silica fume [4], granulated blast furnace slag [5], metallurgical slag [6,7,8,9], etc. Mineral admixtures used in cementitious composites can reduce engineering costs, reduce the heat of hydration, and improve long-term mechanical properties and durability.

Metallurgical slag is a byproduct generated during high-temperature metallurgical processes, and its large quantity and complex chemistry have been a burden and barrier to industrial development [10]. As a byproduct of iron ore, approximately 2.5–3 tons of iron tailings are generated for every ton of iron produced [11,12,13]. For copper tailings, approximately 196.5 tons of tailings are generated for every 1 ton of copper produced [14]. These tailings often contain high concentrations of heavy metals, which could have negative impacts on ecosystems and humans [15,16]. Therefore, the use of these massive volumes of mine tailings could be seen as a step forward in the long-term management of mineral waste. Mo [17] investigated the use of copper tailings as a cementitious material in concrete construction, finding that copper slag could stabilize heavy elements and have acceptable properties. Iron tailings are mainly reported to be used as aggregates in the field of building materials, although the fundamental chemical compositions of iron tailings are similar to those of cement [18,19,20,21].

Copper and zinc tailing powder are finely ground waste after copper minerals and zinc minerals have been extracted from ores during beneficiation. On the one hand, some valuable elements, such as Zn, Fe, Cu, etc., in CZTP can be recycled. On the other hand, according to the chemical components of CZTP as provided in [22], CZTP has a high total silica, alumina, and iron oxides (SiO_2_ + Al_2_O_3_ + Fe_2_O_3_) of more than 71%, indicating that CZTP can potentially be suitable for use as a binding agent and significantly reduce the energy consumption in the cement production process [23]. Partial replacement of ordinary Portland cement with CZTP for preparation of mine tailings-based cemented paste backfill results in a compact microstructure and better pore distribution with lower porosity [24]. The use of an alkali activator improves the compressive strength of composite cementitious materials containing CZTP [25]. Similar to copper slag [26,27] and zinc slag, the use of CZTP as a cementitious material has been reported in the literature, but on a relatively smaller scale. The composite cementitious materials containing copper and zinc tailing powder can be used as mineral admixtures to replace partial Portland cement in the preparation of concrete. The cost of concrete can be significantly reduced. Meanwhile, the emission of carbon dioxide has evidently decreased due to the reduction in the amount of Portland cement in concrete.

To expand the utilization of CZTP, the effects of water content on the properties, including the strength development of cement mortar prepared with different contents of CZTP, were investigated in the present research work. Hydration heat evolution, chemically bonded water, pore structure, hydrates, and microstructure were also studied to clarify the relation between mechanical properties and microstructure.

## 2. Materials and Methods

### 2.1. Materials and Mix Design

CZTP, Portland cement P. I 42.5, ISO standard sand, and potable water were used as raw materials. The main chemical components of CZTP were determined by X-ray fluorescence (XRF) analysis and are given in Table 1. As illustrated in Figure 1, the main mineral compositions of copper and zinc tailing powder are hedenbergite (CaFeSi_2_O_6_) and diopside (CaMgSi_2_O_6_). According to Table 1 and Figure 1, copper and zinc tailing powder contain considerable amounts of SiO_2_ and CaO (accounting for almost 60%), indicating that CZTP has certain potential cementitious properties.

Figure 2 shows the SEM images of the copper-zinc tailing powder. As depicted in Figure 2, most of the particles in copper and zinc tailing powder show irregular blocky, granular, and clastic shapes. The particle size distributions of cement and CZTP were measured by a laser particle size analyzer and the results are depicted in Figure 3. As shown in Figure 3, the specific area of copper and zinc tailing powder is 497.6 m^2^/kg, and the sizes of most particles are between 20 μm and 60 μm. Moreover, the particle size of CZTP is smaller than that of cement (the specific area of Portland cement is 386 m^2^/kg). Therefore, they can penetrate into the pores of the matrix, thereby reducing the porosity of the hardened cement paste.

The formulations to produce composite binder pastes and mortars are shown in Table 2 and Table 3, respectively. A water-to-binder (w/b) ratio of 0.45 was used, and the substitution ratios of CZTP were 0, 10%, 20%, 30%, and 50% by mass, with additions of 7.6 g superplasticizer. The samples given in Table 2 and Table 3 were identified using the selected parameters of w/b and CZTP content. For example, specimen 3_20b indicates a paste produced with 20 wt.% CTZP at w/b of 0.30. Specimen 3_20m indicates a mortar produced with 20 wt.% CZTP at w/b of 0.30. The specimens were demolded after 24 h and then maintained at 20 ± 1 °C and >90% RH.

### 2.2. Test Methods

The hydration calorimetry test was performed at a constant temperature of 25 ± 0.1 °C, and approximately 76 h of data were recorded. The chemically bound water in the hardened paste was determined using the burning method. The CZTP replacement ratios of the composite cementitious pastes prepared are illustrated in Table 2. In total, 200 g of fresh paste was weighed and roughly divided into 10 equal portions before placing in plastic tubes. The specimens were sealed and cured until they reached a specified age (3 d, 7 d, 28 d, and 56 d) at 20 °C. Sample preparation procedures for the characterization of the chemically bound water included soaking for 24 h in ethanol, oven drying for 24 h at 80 °C, crushing and grinding. Then, the ground powder was calcinated in a crucible furnace at 1050 °C for 3 h. The weight difference before and after calcination, as well as the loss upon ignition of the raw materials, corresponds to the chemically bonded water content.

The Chinese standard GB/T 17671-1999 was used as the test method of flexural and compressive strength. Flexural strength and compressive strength were determined from triplicate 40 mm× 40 mm × 160 mm specimens after 3, 7, 28, and 90 days of curing under standard curing conditions. Pastes cured for 3 and 28 days were crushed and subsequently ground using an agate mortar and passed through an 80 μm sieve prior to XRD analysis, and the diffraction patterns were collected between 5 and 60 °C using Cu Kα radiation (tube voltage: 40 kV, tube current: 120 mA) as a source at a rate of 6°/min. In addition, the microstructure and morphology of the reactants were examined by environmental scanning electron microscopy (ESEM, FEI Quanta 250). The porosity was estimated by mercury intrusion porosimetry (MIP). The samples are hardened cement pastes.

## 3. Results and Discussion

### 3.1. Characteristics of Hydration Heat Evolution

Figure 4 shows the hydration heat evolution rate and the cumulative hydration heat of the composite binder with different contents of CZTP and w/b. As depicted in Figure 4a,b, there are two exothermic peaks on the hydration heat evolution rate curve of the binders investigated regardless of w/b. The sharp exothermic peak corresponding to the first peak occurring in the curves immediately after mixing the binder with water is attributed to the quick dissolution of cement. The second exothermic peak mainly is ascribed to the hydration of cement. The rate of the second exothermic peak and the total hydration heat release at different hydration times determined from the heat evolution curves are listed in Table 4. The increase in CZTP content and the w/b decrease in the hydration evolution rate are in accordance with Table 4.

As shown in Figure 4a,b, an increase in the replacement ratio of CZTP prolongs the ending time of the induction period and reduces the second peak value of the hydration heat evolution rate of the composite cementitious system investigated. This is attributed to the decreased amount of Portland cement used, extending the time required for the hydration products to reach supersaturation with increasing CZTP content. In addition, the reactivity of CZTP is lower than that of cement, and the hydration heat evolution rate of CZTP is obviously lower, leading to a decrease in the overall hydration heat evolution rate of the cementitious composite. Furthermore, as shown in Figure 4a,b, when the CZTP percentage is constant, increasing w/b decreases the hydration heat evolution rate of the second

As illustrated in Figure 4c,d, similar to other mineral admixtures [27,28], the cumulative hydration heat decreased as the cement replacement ratio with CZTP or w/b increased. As illustrated in Table 4, for the pastes prepared with a w/b of 0.3, the heat emission at 72 h decreases by 5.8% and 29.3%, respectively, when the content of CZTP increases from 20% to 50%. For paste with a w/b of 0.45, the heat emission at 72 h decreases by almost 10.7%, from 265.3 to 237.0 J/g after the incorporation of 20% CZTP and decreases by approximately 35.5% when the CZTP content increases to 50%.

It is important to note that the heat release per gram of cement of the composite binder clearly increases with increasing CZTP content, as shown in Table 4. This is attributed to the dilution effect of CZTP. When w/b remains constant, increasing the CZTP content causes the water-to-cement ratio in the system to rise, which facilitates the hydration of cement. Therefore, the early-stage hydration rate of cement is accelerated, and the heat release per gram of cement increases.

### 3.2. Strength Development

Figure 5a,b illustrate the flexural and compressive strength development, as well as the strength development rate of cement pastes with various CZTP contents and w/b values of different ages under standard curing conditions. The increase in CZTP content and w/b decrease the flexural and compressive strength of the specimens. This is attributed to the fact that the substitution of cement with CZTP increases the water for the reaction of cement due to the dilution effect, facilitating the reaction of cement to some extent at an early age. However, it also reduces the quantity of cement and thus decreases the overall quantity of the hydration products, leading to the lower strength development of hardened cement paste.

As depicted in Figure 5, the early strength of all samples increased rapidly after mixing; however, the flexural and compressive strength development rates at various ages differed. The strength development rate for samples from 3 d to 7 d decreases initially and subsequently increases and then declines with increasing CZTP content in blended cement with a w/b of 0.3, whereas the strength development rate for specimens from 7 d to 28 d increases with increasing CZTP content. Furthermore, the flexural strength from 3 d to 7 d develops faster than that from 7 d to 28 d with the increase in the substitution ratio of the cement, and the maximum growth rate is achieved when the content of CZTP is 20%. Following that, the flexural strength from 3 d to 7 d develops faster than that from 7 d to 28 d. The compressive strength develops faster from 3 d to 7 d than from 7 d to 28 d when the CZTP content is less than 50%. As the CZTP content is 30%, the strength develops quickly, and when the CZTP content is increased, the strength immediately diminishes. When the CZTP content is at 30%, the compressive strength grows the fastest, but as the CZTP percentage is increased to 50%, the development rate slows dramatically.

When the w/b was increased to 0.45, the early flexural strength and compressive strength of samples on day 7 were 4–12% and 29–51% higher than those on day 3, respectively, exhibiting a considerable increase rate. With increasing CZTP content, the strength development rate for samples from 3 to 7 days first increased, then decreased, and finally increased. The growth rate reached its maximum value when the CZTP content was 50% (flexural strength development rate: 12.8%; compressive strength development rate: 50.8%). With increasing CZTP content, the rate of flexural strength development of specimens from 7 to 28 days dropped at first, then increased, and eventually diminished. The compressive strength development rate of the samples first increases and then drops as the amount of CZTP added increases.

In conclusion, the substitution of cement with CZTP improves the strength growth rate of specimens to some extent, especially the strength development rate from 7 to 28 days. This may be attributed to the fact that as the reaction proceeded, the long-term strength increased gradually, and the strength results indicate that the reaction continues for some time after the final set and would produce a denser matrix with time. However, when the CTZP content is too high, it is detrimental to the strength growth.

### 3.3. Pore Evolution

The cumulative and differential distribution of pore volume of the hardened composite binder with different CZTP contents and at various ages are presented in Figure 6. Figure 6(a-1,b-1,c-1,d-1) present the MIP results expressed by differential intrusion curves of mercury of cement pastes with different contents of CZTP and various w/b, while Figure 6(a-2,b-2,c-2,d-2) illustrate the pore volume content of samples.

The pore size distribution can better reflect the microstructure of hardened cement paste. According to Mehta’s study [29], the pores in hardened cement paste can be classified into the following four grades based on the pore size: <4.5 nm (gel micropores), 4.5 nm–50 nm (mesopores), 50 nm–100 nm (middle capillary pores), and >100 nm (large capillary pores), and he argued that pores larger than 50 nm had a significant effect on the strength and permeability, while pores smaller than 50 nm, which were regarded as microscopic pores, mainly affected drying shrinkage and creep.

As shown in Figure 6(a-1), the critical pore diameter (dc) of 3_0 at 3 d is determined to be 95.3 nm, which is higher than that of 3_20 (50.3 nm) and 3_50 (62.5 nm). This is consistent with the refinement of CZTP’s pore-size effects on pore structure. As illustrated in Figure 6(a-2), the addition of CZTP enhances the pore volume content of the hardened matrix, and the pore volume increases with the CZTP content. It is worth noting that, in addition to the peak corresponding to the critical diameter, there is another peak on the log differential curve from the MIP results for both samples 3_20 and 3_50, and the corresponding pore diameters are 6 nm and 7.2 nm, respectively, indicating that CZTP can help to lower the overall volume of large capillary pores, which are considered to be harmful pores [30] while promoting the formation of gel micropores or mesopores in composite cementitious materials.

As shown in Figure 6(b-1), the critical pore diameters of 4.5_0, 4.5_20, and 4.5_50 cured for 3 d are recorded as 62.5 nm, 95.3 nm, and 480.1 nm, respectively. The critical pore diameters of samples with a w/b of 0.45 increase with the CZTP content. In comparison to 4.5_20 and 4.5_50, the reference sample, 4.5_0, has the highest pore volume content. The pore volume content of sample 4.5_20 is the lowest. CZTP promotes the formation of large capillary pores and gel micropores in samples with higher w/b after curing for 3 d. The increase in w/b mainly improves the volume content of the gel micropores and the large capillary pores of samples cured for 3 d, as shown in Figure 6(b-2).

As shown in Figure 6(c-1), the critical pore diameters of 3_0, 3_20, and 3_50 cured for 28 d are 56.6 nm, 32.4 nm, and 40.3 nm, respectively. The critical pore diameter decreases with increasing CZTP content, while the pore volume content increases with increasing CZTP content, as depicted in Figure 6(c-2). The pore volume content decreases with age.

As illustrated in Figure 6(d-1), the critical pore diameters for 4.5_0, 4.5_20, and 4.5_50 with a w/b of 0.45 after curing for 28 d are 77.1 nm, 62.5 nm, and 120.8 nm, respectively. As depicted in Figure 6(d-2), sample 4.5_0 has the highest pore volume content, while sample 4.5_20 has the lowest pore volume content. CZTP promotes the formation of large capillary pores and gel micropores, and CZTP can significantly reduce the volume of middle capillary pores.

As shown in Figure 6, CZTP decreases the number of large capillary pores while increasing the volume of gel pores in all the samples cured for 3 and 28 d. When w/b is 0.3, the pore volume content increases as the CZTP content increases. When w/b is 0.45, the reference sample has the highest pore volume content, while the sample with 20% CZTP has the lowest pore volume content. The pore volume content of all samples decreases with age.

### 3.4. Chemically Bonded Water

The chemically bonded water can be used to measure the degree of hydration [31]. Figure 7 shows the chemically bonded water of hardened composite paste cured to various ages at ambient temperature. As depicted in Figure 7, the early hydration of pure cement was rapid, and the chemically bonded water was high, while the amount of chemically bonded water in the samples investigated decreased with increasing CZTP content at early ages. The reason for this was that the potential activity of CZTP had not been fully activated and the reaction rate was slow. Although the dilution effect and nucleation effects of CZTP in composite cement paste could improve the hydration degree of cement at an early stage, the lower early reaction degree of CZTP resulted in a lower composite binder reaction degree than pure cement. With increasing CZTP content, the total degree of hydration reaction decreased, leading to a lower early chemical bonding water content of the composite binder.

Compared with cement-slag complex cementitious materials [32], the amount of chemically bonded water in cement-CZTP blended cementitious materials never increased more than that of pure cement hardened paste, which may be attributed to the fact that the activity of slag was higher than that of CZTP, and the activity of CZTP could not be fully activated even at long-age curing.

The chemically bonded water of the composite binder increased as the w/b ratio increased, as seen in Figure 7a,b. On the one hand, the larger the water-binder ratio was, the more equally spread and in contact with water the particles of the cementitious material are; on the other hand, more space can be supplied for the growth of hydration products. As a result, in the sample with a higher w/b, the reaction degree of the cementitious material is higher, more hydration products are formed, and the amount of chemically bonded water is higher.

### 3.5. Powder X-ray Diffraction

Figure 8 depicts the mineral composition of samples cured for 3 d and 28 d with varying amounts of CZTP. The type of hydration product was not affected by the curing age or CZTP. The XRD data confirms the presence of ettringite, portlandite, and calcite in all hydrated samples. The ettringite was considered to be an early hydration product [33], which was also found in the hardened cement paste cured for 3 d and 28 d, albeit only in trace amounts. The detected hedenbergite and diopside in the samples came from CZTP. Unreacted C_3_S can also be detected in the hardened matrix.

### 3.6. Scanning Electron Microscopy

As shown in Figure 7a,b, many pores were detected in the hardened blended cement paste after curing for 3 d. Although some C-S-H gels formed, they were all tiny particles with loose structures. In the hardened matrix that had been cured for 28 days, a large amount of C-S-H gel formed, and the structure was exceedingly compact and dense. lamellar Ca(OH)_2_ interlocked with each other, resulting in a denser structure.

It seemed that increasing the CZTP content had no effect on the microscopic morphology of the hardened pastes cured for 3 d, as depicted in Figure 9c. The C-S-H gel was small and highly dispersed. A large amount of lamellar Ca(OH)_2_ could be detected in the hardened matrix, which closely bonded with the C-S-H. Most CZTP particles were embedded in the C-S-H gel, indicating that CZTP mainly played the role of filling. In addition, a small amount of CZTP was surrounded by hydration products, acting as microaggregates.

The long-term mechanical properties and microstructure, durability, and heavy metal leaching behavior of composite cementitious materials containing copper and zinc tailing powder will be further investigated.

## 4. Conclusions

(1) By adding CZTP to cementitious materials at 25 °C, the exothermic rate and cumulative hydration heat can be lowered, and the higher the dosage is, the greater the reduction.

(2) Both the flexural and compressive strengths of the composite binder decrease with increasing CZTP content and w/b.

(3) The critical pore diameters of the specimens decrease with the curing age. CZTP reduces the critical pore diameter of samples with low w/c (w/b = 0.3) while increasing the pore volume content. CZTP reduces the pore volume content and increases the critical pore diameters in samples with a w/b of 0.45 and cured for 3 d. For samples cured for 28 d, a low CZTP content reduces the critical pore diameter of the specimens, whereas a high CZTP level increases the critical pore diameter. CZTP reduces the content of large capillary pores and increases the volume of gel micropores and middle capillary pores in the hardened cement paste with low w/b. For high w/b, the large capillary pores and gel micropores are increased in the Portland-CZTP system.

(4) The chemically bonded water content of the cementitious composite containing CZTP decreases as the CZTP content increases and increases with the curing age. The amount of chemically bound water grows as w/b increases.

(5) The hydration product type of the blended paste is not changed by CZTP, and the XRD data confirms the presence of ettringite, portlandite, and calcite. CZTP particles mainly play the role of filler, and a small amount acts as a microaggregate.

## Figures and Tables

**Figure 1 materials-15-05612-f001:**
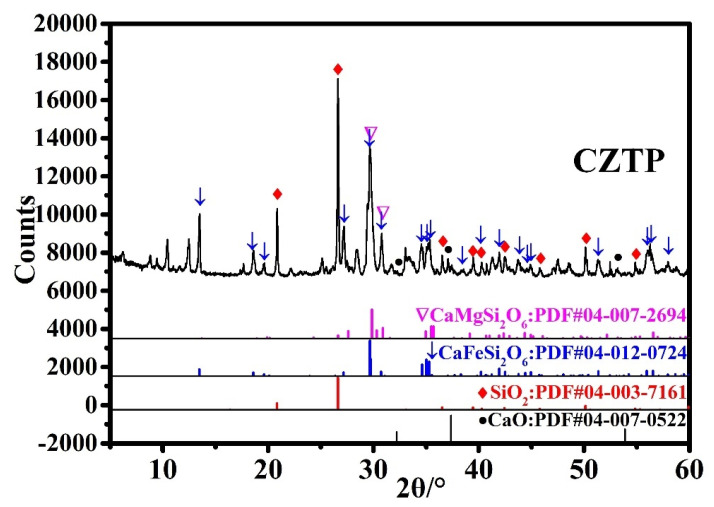
XRD pattern of copper-zinc tailing powder.

**Figure 2 materials-15-05612-f002:**
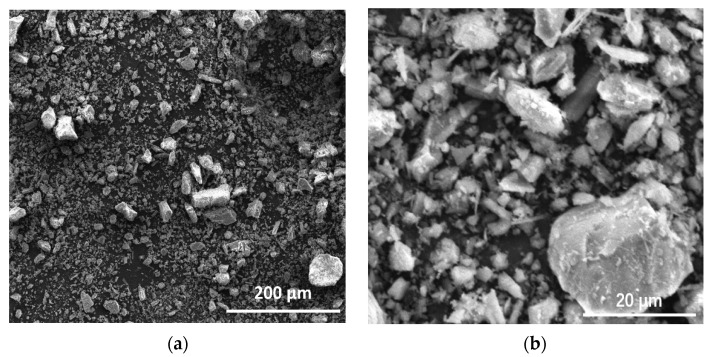
SEM images of copper-zinc tailing powder at low magnification (**a**), and at high magnification (**b**).

**Figure 3 materials-15-05612-f003:**
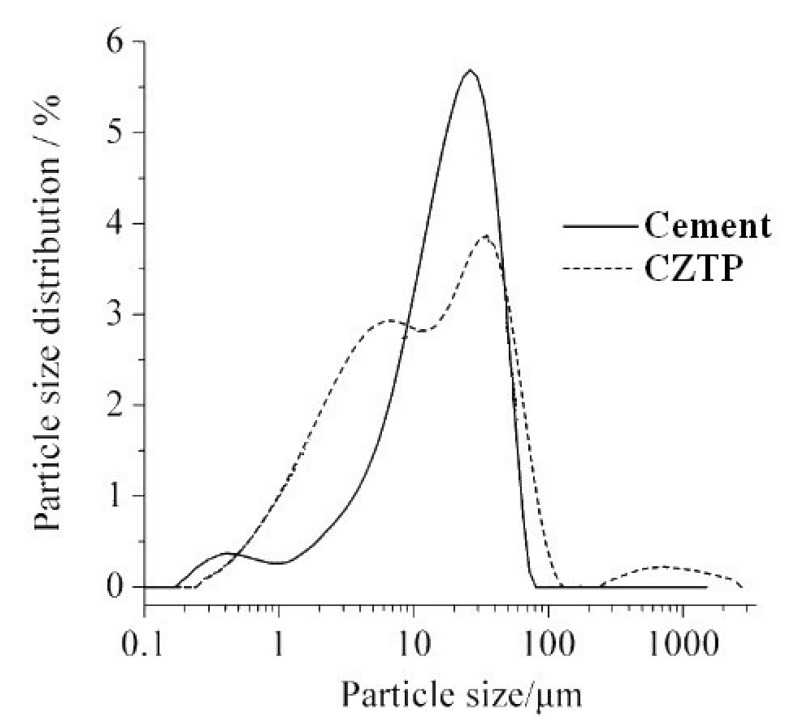
Particle size distribution of Portland cement and copper-zinc tailing powder.

**Figure 4 materials-15-05612-f004:**
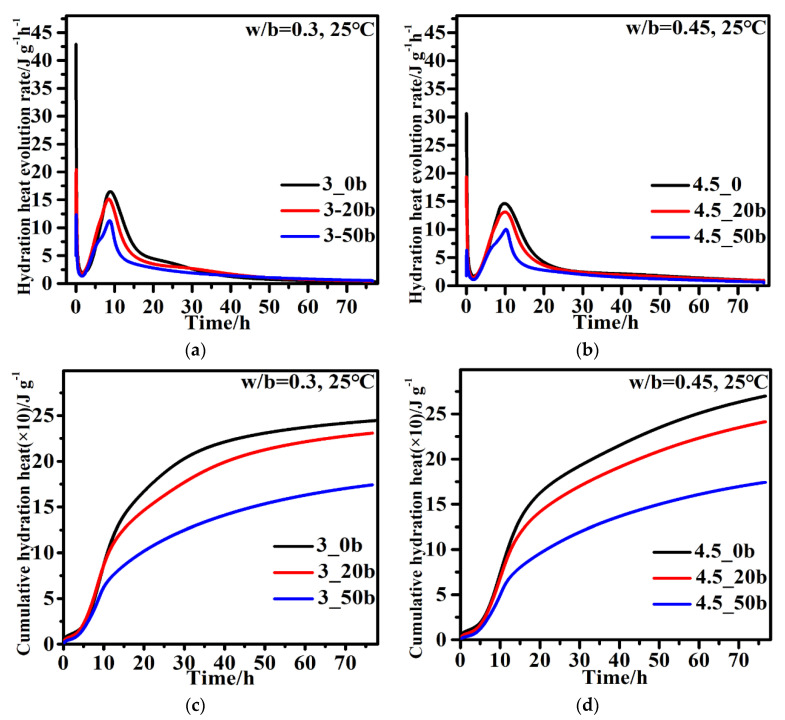
Hydration heat evolution rate: (**a**) w/b = 0.3, (**b**) w/b = 0.45; cumulative hydration heat: (**c**) w/b = 0.3, (**d**) w/b = 0.45.

**Figure 5 materials-15-05612-f005:**
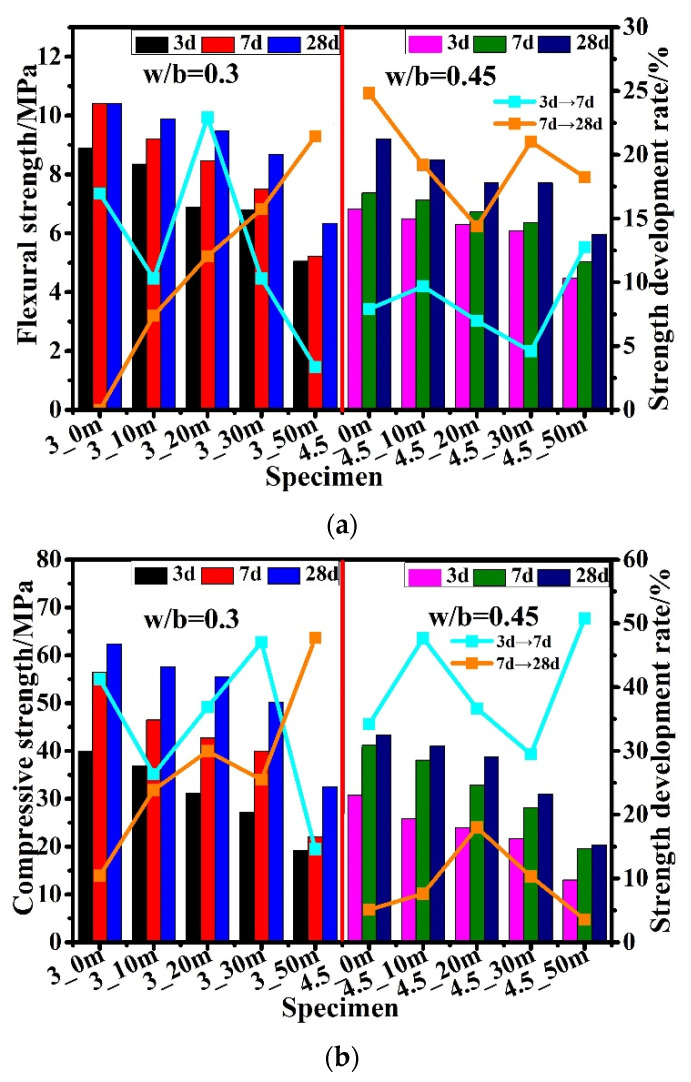
(**a**) Flexural strength; (**b**) compressive strength.

**Figure 6 materials-15-05612-f006:**
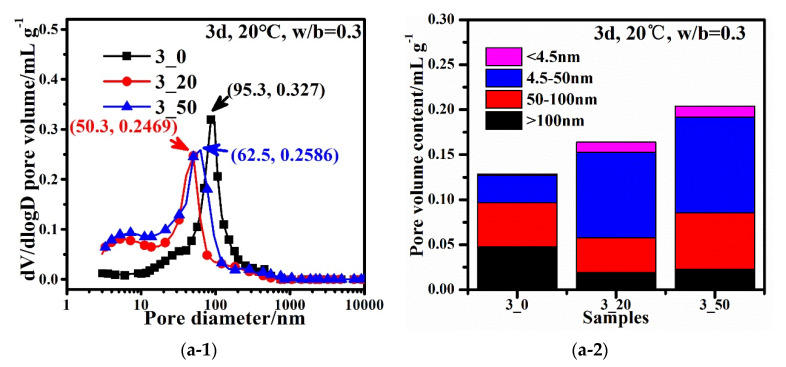
Pore size distribution and pore volume content of samples at different ages: (**a**) samples with a w/b of 0.3 and cured for 3 d, (**a-1**) presents the differential intrusion curves of mercury of cement pastes with different contents of CZTP and various w/b, while (**a-2**) illustrates the pore volume content of corresponding samples (the same with the following (**b-1**,**c-1**,**d-1**) and (**b-2**,**c-2**,**d-2**); (**b**) samples with a w/b of 0.45 and cured for 3 d; (**c**) samples with a w/b of 0.3 and cured for 28 d; (**d**) samples with a w/b of 0.45 and cured for 28 d.

**Figure 7 materials-15-05612-f007:**
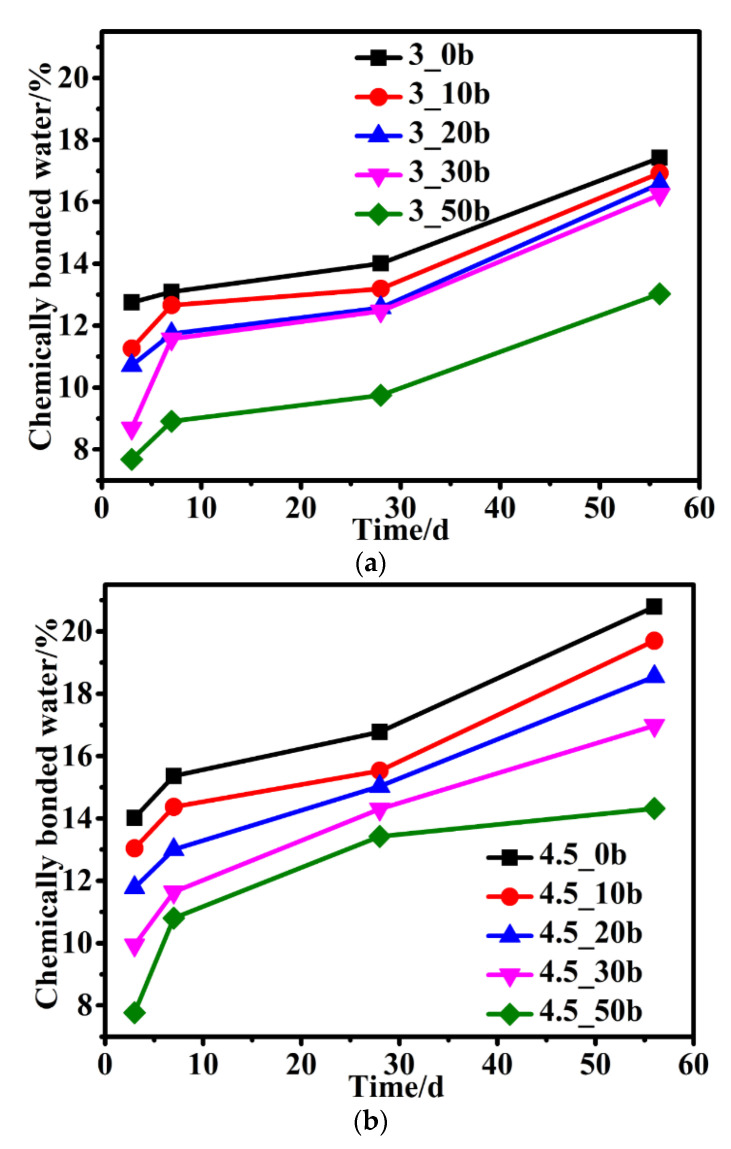
Chemically bonded water of samples at different ages: (**a**) w/b = 0.3, (**b**) w/b = 0.45.

**Figure 8 materials-15-05612-f008:**
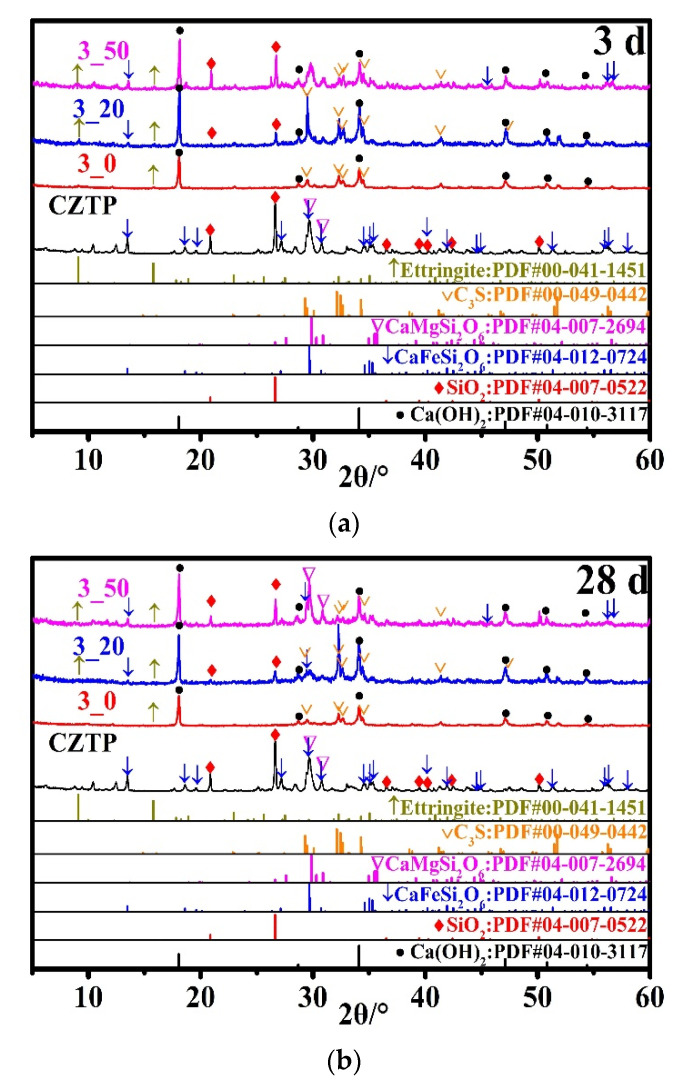
XRD patterns of samples cured for (**a**) 3 d and (**b**) 28 d.

**Figure 9 materials-15-05612-f009:**
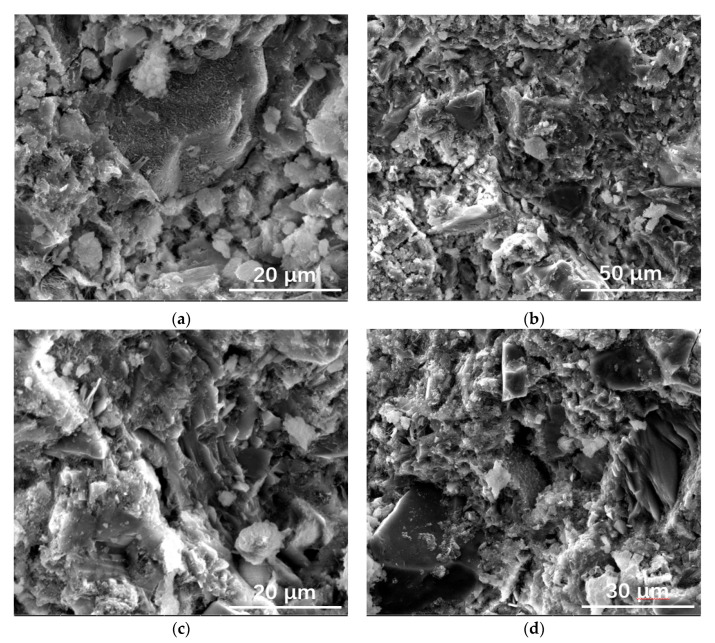
SEM images of samples: (**a**) 3_20 cured for 3 d; (**b**) 3_20 cured for 28 d; (**c**) 3_50 cured for 3 d; (**d**) 3_50 cured for 28 d.

**Table 1 materials-15-05612-t001:** Chemical composition of unmodified and modified copper and zinc tailing powder (wt.%).

Materials	SiO_2_	Fe_2_O_3_	CaO	MnO	MgO	SO_3_	Al_2_O_3_	ZnO	K_2_O	TiO_2_
Cement	20.55	3.27	62.50	-	2.61	2.93	4.59	-	-	-
CZTP	37.10	22.16	19.90	6.93	1.26	5.47	4.74	0.73	0.59	0.49

**Table 2 materials-15-05612-t002:** Mix proportions of composite binder pastes.

Sample ID	w/b	Binder Composition/%	
		Cement	CZTP
3_0b	0.30	100	0
3_10b		90	10
3_20b		80	20
3_30b		70	30
3_50b		50	50
4.5_0b	0.45	100	0
4.5_10b		90	10
4.5_20b		80	20
4.5_30b		70	30
4.5_50b		50	50

**Table 3 materials-15-05612-t003:** Mix proportions of mortars.

Sample ID	w/b	Binder Composition/%		Standard Sand/g	Superplasticizer/g
		Cement	CZTP		
3_0m	0.30	450	0	1350	10.8
3_10m		405	45		
3_20m		360	90		
3_30m		315	135		
3_50m		225	225		
4.5_0m	0.45	100	0	1350	7.6
4.5_10m		90	10		
4.5_20m		80	20		
4.5_30m		70	30		
4.5_50m		50	50		

**Table 4 materials-15-05612-t004:** Characteristic values of the hydration heat evolution curves of specimens at 25 °C.

Sample	Rate of the Second Heat Emission Peak q_max_ (J/g·h)	Total Heat Release (J/g)				Heat Release per Gram of Cement			
		12 h	48 h	60 h	72 h	12 h	48 h	60 h	72 h
3_0b	16.484	114.2	229.3	237.7	242.6	114.2	229.3	237.7	242.6
3_20b	15.155	107.1	210.4	221.3	228.6	133.9	263.0	276.6	285.8
3_50b	11.268	74.1	151.3	162.9	171.5	148.2	302.6	325.8	343.0
4.5_0b	14.606	101.7	231.2	250.7	265.3	101.7	231.2	250.7	265.3
4.5_20b	13.107	93.1	205.5	223.3	237.0	116.4	256.9	279.1	296.3
4.5_50b	10.020	66.5	147.5	160.9	171.1	133.0	295.0	321.8	342.2

## Data Availability

Not applicable.

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
