# Peer review of "Influence of Copper and Zinc Tailing Powder on the Hydration of Composite Cementitious Materials"

_materials, 2022, doi:10.3390/ma15165612_

Round 1

Reviewer 1 Report

I suggest improving it by adding the following data:

-        what is the possibility of using the analyzed composite cementitious materials in practice, what makes its use difficult (technological and economic aspects, or so on), what are the complicating and facilitating factors),

-        what are the effects of the analyzed composite cementitious materials on reinforcement in concrete

-        whether and when durability will be tested,

-        What are the suggestions for further research into using concrete with the addition of?

Author Response

 (1) Comment: what is the possibility of using the analyzed composite cementitious materials in practice, what makes its use difficult (technological and economic aspects, or so on), what are the complicating and facilitating factors)?

Response: The analyzed composite cementitious materials containing copper and zinc tailing powder can be used as mineral admixtures to replace partial Portland cement in preparation of concrete. The cost of concrete can be significantly reduced. Meanwhile, the emission of carbon dioxide is evidently decreased due to the reduction in the amount of Portland cement in concrete.

  We think the difficulty of utilization of copper and zinc tailing powder is the heavy metal leaching behavior of copper and zinc tailing powder. To ensure the environment safety, the heavy metal leaching behavior must be investigated. We have been doing this research. The research results will be published in the future.

(2) Comment: what are the effects of the analyzed composite cementitious materials on reinforcement in concrete?

Response: We are so sorry that the effects of the analyzed composite cementitious materials on reinforcement in concrete are not investigated. The reasons are as follows:

(1) The Cl was not found in the copper and zinc tailing powder according to the XRF results. Thus, the addition of copper and zinc tailing powder doesn’t affect the properties of reinforcement.

(2) The replacement ratio of copper and zinc tailing powder is smaller than 50% in this study. The mass fraction of Portland cement is more than 50% in the composite cementitious materials. The pH value of pore solution in composite cementitious materials containing copper and zinc tailing powder is still much high based on our previous research results. Therefore, the high alkalinity of composite cementitious materials containing copper and zinc tailing powder can provide protection for the reinforcement.

(3) Comment: whether and when durability will be tested?

Response: The durability of composite cementitious materials containing copper and zinc tailing powder has been investigated, such as resistance to the penetration of chloride ions, resistance to sulfate attack, resistance to carbonation, et al. Another manuscript related to the results of durability will be wrote and submitted to the Journal in the future.

(4) Comment: What are the suggestions for further research into using concrete with the addition of?

Response: The long-term mechanical properties and microstructure, durability and heavy metal leaching behavior of concrete containing copper and zinc tailing powder are suggested for further research.

Reviewer 2 Report

Journal: Materials (ISSN 1996-1944)

Manuscript ID: materials-1807071

Type: Article

Title: Influence of copper and zinc tailing powder on the hydration of composite cementitious materials.

Authors: Weiwei Han, Fanghui Han*, Ke Zhang.

a)           Keywords, write 5 minimum.

b)          Introduction: In the literature, add more than three to the number of authors who have worked on copper and zinc tailing powder and achieved results.

c)           Figure 1 revise 2 theta (o) and put legend.

d)          Figure 2, SEM image, the particles are not clear, put the zoom on the region required.

e)           For references, refer to these refs. Distribution of the particles using TEM and SEM images

DOI: https://doi.org/10.1098/rsos.172214

DOI: https://doi.org/10.1007/s10854-017-7901-7

Best Regards

--------------------------------------------------------

Author Response

(1) Comment: Keywords, write 5 minimum.

Response: Five keywords have been given in the revised manuscript.

(2) Comment: Introduction: In the literature, add more than three to the number of authors who have worked on copper and zinc tailing powder and achieved results.

Response: Literature [22], [23], [24] and [25] are related to the study on copper and zinc tailing powder. The achieved results in the  literature have been given in the revised manuscript. Due to the limited research on effect of copper and tailing powder on properties of cementitious materials, a small amount of literature is found.

(3) Comment: Figure 1 revise 2 theta (o) and put legend.

Response: We have checked the Figure 1. We think the Figure 1 is right. The legend has been given in Figure 1.

(4) Comment: Figure 2, SEM image, the particles are not clear, put the zoom on the region required.

Response: Figure 2(a) is the picture obtained at low magnification. Figure 2(b) is the picture obtained at high magnification. The particle size and morphology can be clearly seen in the Figure 2(b).

(5) Comment: For references, refer to these refs. Distribution of the particles using TEM and SEM images

DOI: https://doi.org/10.1098/rsos.172214

DOI: https://doi.org/10.1007/s10854-017-7901-7

Response: Thank you very much for your suggestions. We usually use SEM to investigate the distribution of particles in cementitious materials. We will use TEM to analyze the distribution of particles in the future.

Reviewer 3 Report

- Line 89. In Table 1 please add the chemical composition for the Portland cement; - Line 102. In Table 2, please add the column "Sample ID", and indicate there the coded designations of the experimental mixes that will display the content of CZTP and w/b; - Lines 162, 164. What do "b" and "m" stand for in coded mixes in all Figures and in Tables? Do they identify "binder" and "mortar" types of mixes? Please add a description to the text. - Line 164. Replace "Table 2" with "Table 4"; - Lines 168-171. The following sentences mention that "The increase in CZTP content and w/b decreases the flexural and compressive strength of the specimens. This is attributed to the fact that the substitution of cement with CZTP increases the water for the reaction due to the dilution effect, facilitating the reaction to some extent at an early age". However, according to Figure 4, an increase in CZTP and w/b reduces the exothermic effect and, as a consequence, the intensity of the chemical reaction. The text description is contrary to Figure 4; - Line 204. In the sentence "However, when the fly ash content is too high, it is detrimental to the strength growth" the authors mention "fly ash content". It should probably be "CZTP content" here; - Line 255. There is a discrepancy between the captions on the pictures in Figure 6 and the main caption for Figure 6. Please check the description of each set in Figure 6 (a, b, c and d).

Author Response

(1) Comment: Line 89. In Table 1 please add the chemical composition for the Portland cement;

Response: The chemical compositions of Portland cement have been added in Table 1 in the revised manuscript.

(2) Comment: Line 102. In Table 2, please add the column "Sample ID", and indicate there the coded designations of the experimental mixes that will display the content of CZTP and w/b;

Response: Thank you for your comment. The column “Sample ID” has been added in the Tables 2 and 3. The coded designations of the experimental mixes have been indicated in the revised manuscript.

 “The samples given in Table 2 and Table 3 were identified using the selected parameters of w/b and CZTP content. For example, specimen 3_20b indicates a paste produced with 20 wt.% CTZP at w/b of 0.30. Specimen 3_20m indicates a mortar produced with 20 wt.% CZTP at w/b of 0.30.”

(3) Comment: Lines 162, 164. What do "b" and "m" stand for in coded mixes in all Figures and in Tables? Do they identify "binder" and "mortar" types of mixes? Please add a description to the text. Line 164. Replace "Table 2" with "Table 4";

Response: The “b” and “m” stand for “binder” and “mortar”, respectively. The meanings of “b” and “m” have been given in the part of “Materials and Methods” in the revised manuscript.

  We are so sorry for our carelessness. The “Table 2” has been replaced with “Table 4” in the revised manuscript.

(4) Comment: Lines 168-171. The following sentences mention that "The increase in CZTP content and w/b decreases the flexural and compressive strength of the specimens. This is attributed to the fact that the substitution of cement with CZTP increases the water for the reaction due to the dilution effect, facilitating the reaction to some extent at an early age". However, according to Figure 4, an increase in CZTP and w/b reduces the exothermic effect and, as a consequence, the intensity of the chemical reaction. The text description is contrary to Figure 4;

Response: It is not contrary to the results of Figure 4. The addition of CZTP accelerates the early reaction of cement due to the dilution effect. According to the results of Table 4, it is clear that the heat release per gram of cement of composite cementitious materials containing CZTP is higher than that of pure Portland cement. However, the reduction in Portland cement plays an important role, so the addition of CZTP decreases the strength.

The sentences have been revised as: “This is attributed to the fact that the substitution of cement with CZTP increases the water for the reaction of cement due to the dilution effect, facilitating the reaction of cement to some extent at an early age”.

(5) Comment: Line 204. In the sentence "However, when the fly ash content is too high, it is detrimental to the strength growth" the authors mention "fly ash content". It should probably be "CZTP content" here;

Response: The “fly content” has been revised as “CZTP content” in the revised manuscript.

(6) Comment: Line 255. There is a discrepancy between the captions on the pictures in Figure 6 and the main caption for Figure 6. Please check the description of each set in Figure 6 (a, b, c and d).

Response: We have checked the captions of pictures in Figure 6. The captions have been revised as “Figure 6. Pore size distribution and pore volume content of samples at different ages: (a) samples with a w/b of 0.3 and cured for 3 d; (b) samples with a w/b of 0.45 and cured for 3 d; (c) samples with a w/b of 0.3 and cured for 28 d; (d) samples with a w/b of 0.45 and cured for 28 d.”